# Computational screen-out strategy for electrically pumped organic laser materials

Qi Ou [1], Qian Peng [2] & Zhigang Shuai [1✉]

Electrically pumped organic lasing is one of the most challenging issues in organic optoelectronics. We present a systematic theoretical investigation to screen out electrical pumping lasing molecules over a wide range of organic materials. With the electronic structure information obtained from time-dependent density functional theory, we calculate multiple photophysical parameters of a set of optical pumping organic laser molecules in our self-developed molecular material property prediction package (MOMAP) to judge whether the electrically pumped lasing conditions can be satisfied, namely, to avoid reabsorption from excitons and/or polarons, and the accumulation of triplet excitons. In addition, a large oscillator strength of $S_1$ and weak intermolecular $\pi$–$\pi$ interaction are preferred. With these criteria, we are able to conclude that BP3T, BSBCz, and CzPVSBF compounds are promising candidates for electrically pumped lasing, and the proposed computational strategy could serve as a general protocol for molecular design of organic lasing materials.

[1] MOE Key Laboratory of Organic OptoElectronics and Molecular Engineering, Department of Chemistry, Tsinghua University, Beijing 100084, China. [2] CAS Key Laboratory of Organic Solids, Institute of Chemistry of the Chinese Academy of Sciences, Zhonguancun Beiyijie 2, 100190 Beijing, China. ✉email: zgshuai@tsinghua.edu.cn

Over the years, great efforts have been devoted to the development of organic solid-state lasers (OSSL) due to the easy and economic fabrication, covering a wide-range lasing wavelength from near infrared to ultraviolet[1–6]. To date, optically pumped OSSL have been maturely developed, with the achievement of remarkably low laser and/or amplified spontaneous emission (ASE) thresholds and quasi-continuous wave operation[7,8]. The ultimate goal in the field of OSSL is, however, to realize electrically pumped lasing so as to truly enable the creation of low-cost, portable, and flexible solid OSSL devices.

Compared to optically pumped OSSL, lasing under electrical pumping is much more challenging because of the following factors. First, triplet exciton tends to accumulate under electrical pumping based on spin statistics[9], that is, three quarters of excitons formed under current injection are triplets, which usually have a long decay time[5]. Second, under electrical pumping, multiple annihilation and absorption losses will be induced by polarons, excitons, and other species that are not involved in optical pumping[5,6]. Recently, there are encouraging progresses reporting promising indications of current injection OSSL[10], which opened up the opportunities in realizing compact and low-cost electrically driven organic laser devices. Notwithstanding the advancements achieved in ref. [10], obstacles still remain for organic laser gain media to observe light amplification under electrical pumping. Therefore, it is of significant importance to carry out theoretical evaluation and predict potentially good electrical pumping candidates, which require a large stimulated emission cross section, minimal annihilation or absorption losses, and a short $T_1$ lifetime, preferably with a high mobility[5,6].

With that in mind, the goal of this communication is to propose a general computational protocol to systematically screen out electrically pumped lasing molecules over a wide range of organic solid-state fluorescent materials, with the assistance of efficient electronic structure calculations based on density functional theory (DFT) and time-dependent DFT (TDDFT). The ability of realizing general lasing behavior will be evaluated from two perspectives that are closely related to the stimulated emission cross section: i.e., the emission oscillator strength and the strength of the intermolecular π–π interaction for π-stacking single-crystal materials. A systematic screening will then be performed for good electrical pumping candidates over 12 optical pumping materials with low laser/ASE thresholds, which focuses on the following criteria: no strong absorption or annihilation among excitons and polarons near the $S_1$ emission wavelength, relatively slow intersystem crossing (ISC) rate from $S_1$ to triplets compared to the radiative rate of $S_1$, and short lifetime of $T_1$. With these criteria and the photophysical parameters obtained from our self-developed molecular material property prediction package MOMAP[11–13], we are able to screen out three candidates with great potential in realizing electrical pumping laser, i.e., BP3T, CzPVSBF, and BSBCz. It should be noted that the lasing/ASE behavior focused in this paper is constrained to the conventional light amplification mechanism that requires a population inversion in organic laser dyes[5]. Other line-narrowing mechanisms, such as polariton-induced lasing[14] and stimulated resonance Raman scattering[15,16], where the threshold is not limited by population inversion, would require different theoretical frameworks to investigate.

## Results

### Investigated systems and predicted emission energies.
The organic fluorescent solid-state materials that will be investigated here are divided into two groups: single-crystalline materials shown in Fig. 1 and thin film materials shown in Fig. 2. Both are selected from experiments published in literature: most of the

materials are chosen from ref. [6] for group I and from ref. [5] for group II, together with a few up-to-date reported materials. The photoluminescence (PL) properties as well as the lasing behavior (if any) of these materials have been experimentally well-demonstrated, which provides a reliable benchmark for our theoretical calculations.

The TDDFT predicted emission energy is compared with experimental values for all materials of interest in Table 1. As shown in Table 1, the absolute error predicted by theoretical calculations is consistently very small. The largest deviation for group I is 0.273 eV from bMODBDCS, while the one for group II is −0.178 eV from CzPVSBF. Both these two deviations are within the reliable deviation range of TDDFT method (~0.3 eV). Furthermore, the mean absolute error (MAE) is 0.105 eV for group I and even smaller for group II. Altogether, these accurate predictions on the emission energy indicate the rationality of our applied electronic structure methods to the fluorescent organics investigated throughout this work.

**Emission oscillator strength and light amplification.** As an important parameter in a laser gain medium, the stimulated emission cross section $\sigma_{em}$ of a laser transition can affect laser performance in terms of threshold energy, output energy, maximum gain, etc. Under conventional lasing mechanism, a large $\sigma_{em}$ is a prerequisite of a good laser gain medium[17]. Theoretically, for a given PL material, $\sigma_{em}$ is directly proportional to the emission oscillator strength $f_{em}$ via[17]

$$\sigma_{em}(\nu) = \frac{e^2}{4\varepsilon_0 m_e c_0 n_F} g(\nu) f_{em}, \qquad (1)$$

where $e$ the electron charge, $\varepsilon_0$ the vacuum permittivity, $m_e$ the mass of electron, $c_0$ the speed of light, $n_F$ the refractive index of the gain material, $\nu$ the frequency of the corresponding emission, and $g(\nu)$ the normalized line shape function with $\int g(\nu) d\nu = 1$. In practice, $f_{em}$ can be obtained intuitively from an electronic structure calculation, and thus serve as a quick screening tool for optical gain materials. We plot $f_{em}$ of all investigated materials in Fig. 3. It is clear in Fig. 3 that four molecules have significantly smaller $f_{em}$ (≤0.3) than other molecules, i.e., DPA, DPEA, DPASBF, and αNPD. Indeed, no light amplification has been observed for these four materials to the best of our knowledge, at least at room temperature. To further highlight the role played by $f_{em}$ in light amplification of the investigated systems, we present the explicit numbers of $f_{em}$ together with the reorganization energy calculated from the adiabatic potential energy surfaces ($\lambda_{aps}$) and experimental PL quantum yield (PLQY) in Supplementary Table 2. As shown Supplementary Table 2, the reorganization energies of all investigated molecules range from 2000 to 5000 cm$^{-1}$, which confirms the fact that an effective four-energy level required by population inversion can be formed and the self-absorption problem is unlikely a severe issue among these systems, including the four non-lasing molecules. In addition, the PLQYs of these four materials are not significantly lower than other materials. In fact, some lasing materials, such as TPDSB have a much smaller PLQY (4–8%) compared to these four materials, which is resulted from a rather large reorganization energy (5627 cm$^{-1}$) that gives rise to excess nonradiative vibrational losses. Nevertheless, compared to low-efficient lasing material TPDSB, these four non-lasing molecules have much smaller $f_{em}$, which directly leads to small $\sigma_{em}$, and therefore any of these four materials is merely possible to serve as a good optical gain medium in practice.

Even though a large $f_{em}$ does not guarantee a lasing behavior (e.g., αDBDCS), what we can conclude here is that it would be quite difficult to observe lasing behavior from $S_1$ with an

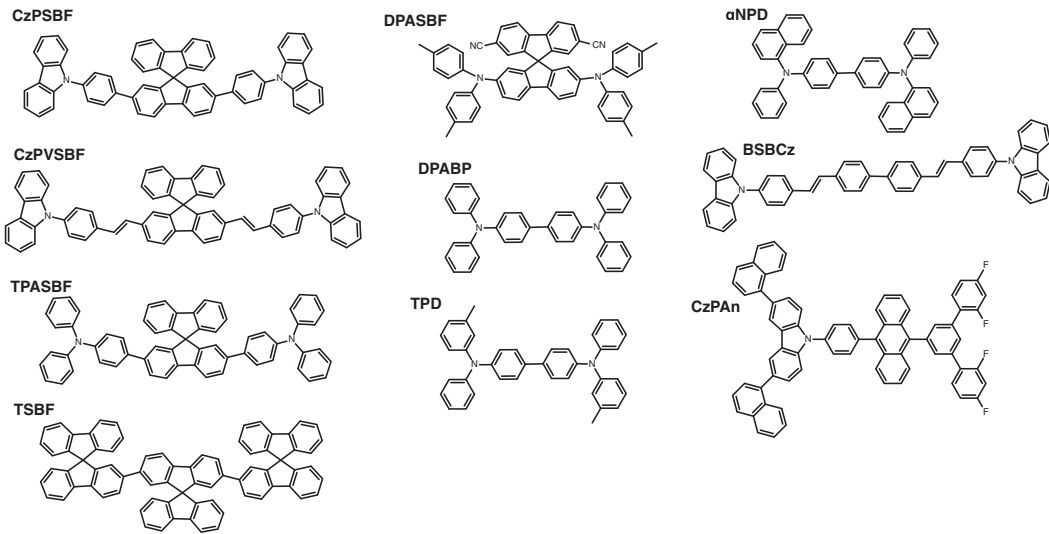

**Fig. 1 Group I: single-crystal organic fluorescent materials investigated in this work.** Most of these single-crystal materials are chosen from ref. [6], with a few up-to-date reported materials.

**Fig. 2 Group II: thin film organic fluorescent materials investigated in this work.** Most of these thin film materials are chosen from ref. [5], with a few up-to-date reported materials.

extremely small $f_{em}$, such as a pure charge-transfer (CT) state where a clear spatial separation is observed between transition orbitals. This explains the fact that most conventional thermally activated delayed fluorescence (TADF) molecules, whose $S_1$ is dominated by a pair of almost orthogonal transition orbitals, are not good laser gain media[5]. Instead, these materials can serve as triplet harvesters to improve the performance of the organic laser dye[18]. With that being said, a few studies have reported lasing activities from newly designed TADF molecules[19–22]. In these studies, the $S_1$ states of the corresponding TADF molecules show effective overlaps (instead of being fully orthogonal) between natural transition orbitals based on the computational results, which potentially enlarge $f_{em}$, and thus will not violate the statement we have drawn here.

**π–π interaction and light amplification in π-stacking system.** Besides the small emission oscillator strength, another adverse factor to light amplification, which has been demonstrated via previous experimental works, is the strong intermolecular π–π interaction in π-stacking single crystals[6,23,24]. Based on the references cited in Table 1, all π-stacking single crystals involved in this study are listed in Table 2, with the corresponding dispersion energies $E_{disp}$ and total interaction energies $E_{tot}$ computed from XSAPT + MBD (many-body dispersion) method, which corresponds to an extended version of the SAPT (symmetry-adapted perturbation theory) that incorporates the dispersion energy evaluated from MBD method[25]. A detailed energy decomposition is listed in Supplementary Table 3. Noting that the interaction energy is proportional to the size of the

**Table 1 Comparison between the computational and experimental emission energies of investigated materials[a].**

| Molecule | $E_{em}^{cal}$ | $E_{em}^{exp}$ | $E_{em}^{cal} - E_{em}^{exp}$ | Refs. | Molecule | $E_{em}^{cal}$ | $E_{em}^{exp}$ | $E_{em}^{cal} - E_{em}^{exp}$ | Refs. |
|---|---|---|---|---|---|---|---|---|---|
| Molecule group I (experimental single-crystal materials) | | | | | | | | | |
| DSB(2PV) | 2.833 | 2.627 | 0.206 | 30 | βDBDCS | 2.405 | 2.532 | −0.127 | 51 |
| 3PV | 2.472 | 2.357 | 0.115 | 30 | αMODBDCS | 2.517 | 2.288 | 0.229 | 23 |
| pMDSB | 2.782 | 2.690 | 0.092 | 30,31 | βMODBDCS | 2.257 | 1.984 | 0.273 | 23 |
| oMDSB | 2.742 | 2.743 | −0.001 | 31 | βPDCS | 2.557 | 2.638 | −0.081 | 52 |
| 4mDSB | 2.830 | 2.649 | 0.181 | 53 | βTFDCS | 2.615 | 2.621 | −0.006 | 29,24 |
| CNMODSB | 2.464 | 2.326 | 0.138 | 54 | DPA | 2.653 | 2.646 | 0.007 | 55 |
| MSMODSB | 2.501 | 2.475 | 0.026 | 56 | DPEA | 2.676 | 2.695 | −0.019 | 57 |
| PDSB | 2.772 | 2.743 | 0.029 | 58 | TPDSB[b] | 2.070 | 2.089 | −0.019 | 59 |
| BMSA | 2.169 | 2.339 | −0.170 | 60 | AC5 | 2.577 | 2.403 | 0.174 | 61 |
| βDCS | 2.597 | 2.572 | 0.025 | 62 | BP3T | 2.194 | 2.168 | 0.026 | 63 |
| αMODCS | 2.488 | 2.500 | −0.012 | 23 | P6T | 1.905 | 1.802 | 0.103 | 64 |
| βMODCS | 2.355 | 2.153 | 0.202 | 23 | TPB | 2.631 | 2.883 | −0.253 | 65 |
| αDBDCS | 2.598 | 2.707 | −0.109 | 43 | | | | | |
| MAE = 0.105 eV | | | | | | | | | |
| Molecule group II (experimental thin film materials) | | | | | | | | | |
| CzPSBF | 2.872 | 2.924 | −0.052 | 66 | αNPD[*c] | 2.845 | 2.786 | 0.058 | 67 |
| CzPVSBF | 2.449 | 2.627 | −0.178 | 7 | DPABP | 2.918 | 2.931 | −0.013 | 67 |
| TSBF | 2.834 | 2.883 | −0.049 | 68 | TPD | 2.901 | 2.924 | −0.023 | 67 |
| TPASBF | 2.709 | 2.749 | −0.040 | 66 | BSBCz | 2.472 | 2.583 | −0.111 | 8 |
| DPASBF[*] | 2.290 | 2.254 | 0.036 | 69 | CzPAn[*] | 2.672 | 2.725 | −0.053 | 70 |
| MAE = 0.061 eV | | | | | | | | | |

[a]The experimental value for each material corresponds to the 0–1 peak of PL spectra measured at room temperature in the cited references, which is usually the strongest peak and the one being amplified in under the lasing behavior (if any) of the corresponding material. Note that these values are taken from undoped materials (either single crystal or thin film) to rule out the influence of the host materials. All energetics are shown in the unit of eV.
[b]Two single-crystalline polymorphs with two slightly different emission wavelengths have been reported. An average emission wavelength is taken here.
[c]Molecules with asterisk are evaluated with an optimal-tuning LRC-ωPBE functional, while others are evaluated with B3LYP functional, the optimal-tuning parameter are listed in Supplementary Table 1. The same applies for Fig. 3.

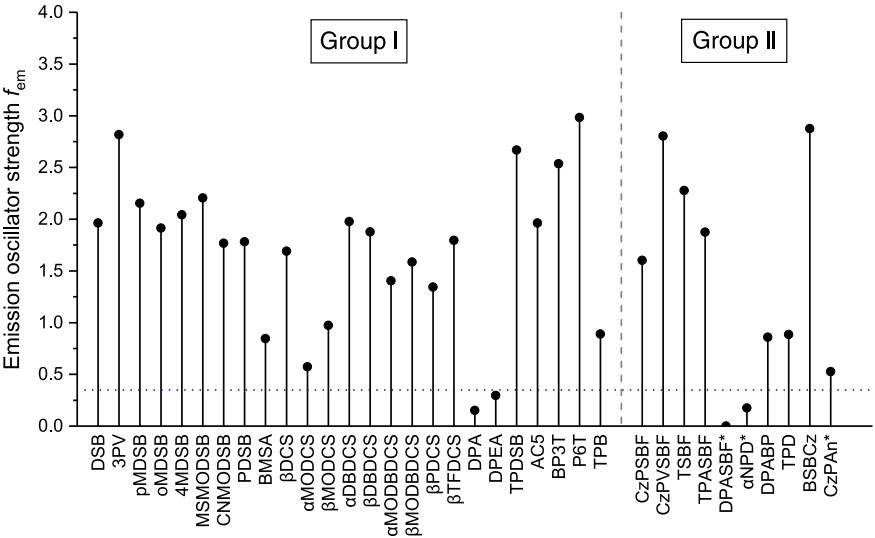

**Fig. 3 Emission oscillator strengths $f_{em}$ of all investigated materials.** Molecules with asterisk are evaluated with an optimal-tuning LRC-ωPBE functional, while others are evaluated with B3LYP functional. The blue dotted line denotes $f_{em} = 0.35$. Four non-lasing materials, i.e., DPA, DPEA, DPASBF, and αNPD, have significantly smaller $f_{em}$.

π-conjugated system[26–28], we have also calculated $E_{tot}$ per π-electron for a fairer comparison. It can be seen from Table 2 that materials without lasing behavior have larger absolute value of $E_{disp}$, which corresponds to a stronger π-stacking effect. Being a major contribution to the attractive forces, the larger absolute value of $E_{disp}$ also leads to a more attractive $E_{tot}$ experienced by each π-electron for non-lasing materials. This is consistent with the experimental fact that these materials are prone to have a broadened excimer-like PL spectra[23,29], accompanied with a smaller emission oscillator strength. The broadened line shape gives rise to a smaller $\sigma_{em}$ at the peak frequency, which may be severe enough to quench light amplification. On the contrary, other π-stacking materials with smaller absolute value of $E_{disp}$ (and thus smaller $E_{tot}$) acquire more slipped π-stacking arrangements, which considerably decrease the intermolecular π–π interaction. Similar results can be obtained from non-π-stacking crystals. pMDSB, for instance, which acquires a her-ringbone arrangement[30,31], has a total interaction energy −14.25 KCal mol$^{-1}$ between its nearest monomer. Due to the smaller interaction energy, excimers are not likely to form inside these

lasing materials and the resulting line shape of the PL spectra is narrower, which gives a larger $\sigma_{em}$ at the peak frequency and thus benefits light amplification.

**Screening out electrically pumped lasing candidates**. To realize electrical pumping lasers, a low threshold in optical pumping is required in addition to the two aforementioned factors, because only one quarter of the excitons formed under current injection are singlets. Among all investigated systems in this work, we choose 12 materials (listed in Table 3) with experimentally low laser/ASE thresholds (as well as low threshold in terms of optical pumping power), and evaluate the possibility of realizing electrical pumping laser based on these materials. The main losses in electrical pumping fluorescence laser are introduced via the following factors[32]. First, the $S_0 \rightarrow S_1$ self-absorption and photo-induced absorptions at the emission wavelength among excitons and polarons to higher excited states ($S_1 \rightarrow S_n$, $T_1 \rightarrow T_n$, and $D_0^{+/-} \rightarrow D_n^{+/-}$, where $D^{+/-}$ denotes polarons) will decrease the effective stimulated emission cross section of $S_1$. Second, singlet exciton may annihilate with singlet/triplet excitons and polarons, which are mainly led by fluorescence resonance energy transfer processes. Third, the ISC from $S_1$ to triplet states will decrease the population of $S_1$. Again, a large stimulated emission cross section is required to overcome various absorption/annihilation losses. Besides, a short lifetime of $T_1$ is also favored to prevent its

accumulation, since the accumulation of $T_1$ will potentially aggravate the aforementioned triplet losses and may lead to thermal degradation of the material[5].

We start with the evaluation of the stimulated emission cross section of $S_1$ according to Eq. (1). For convenience, we rewrite $\sigma_{em}$ as a function of wavelength by inserting $g(\nu) = \frac{g(\lambda)d\lambda}{d\nu} = g(\lambda)\frac{\lambda^2}{c_0}$ into Eq. (1)[17].

$$\sigma_{em}(\lambda) = \frac{e^2\lambda^2}{4\varepsilon_0 m_e c_0^2 n_F} g(\lambda) f_{em}, \qquad (2)$$

where is $g(\lambda)$ the normalized line shape function expressed in the wavelength domain with $\int g(\lambda)d\lambda = 1$. A universal Gaussian broadening with a 75 nm full-width at half-maximum (FWHM) is applied to all investigated systems. The refractive index for crystal and thin film materials are set to be 2 and 1.5, respectively. Secondly, we examine the existence of strong absorption in the vicinity of $S_1$ emission wavelength, represented by the absorption cross section between $S_0/S_1$, $S_1/S_n$, $T_1/T_n$, and $D_0^{+/-}/D_n^{+/-}$. Similar to the emission cross section, the absorption cross section is calculated via[33]

$$\sigma_{abs}^{X_i \rightarrow X_j}(\lambda) = \frac{e^2\lambda^2}{4\varepsilon_0 m_e c_0^2 n_F} g(\lambda) f_{abs}^{X_i \rightarrow X_j}, \qquad (3)$$

where $f_{abs}^{X_i \rightarrow X_j}$ corresponds to various absorption oscillator strengths ($X_i \rightarrow X_j = S_0 \rightarrow S_1$, $S_1 \rightarrow S_n$, $T_1 \rightarrow T_n$, and $D_0^{+/-} \rightarrow D_n^{+/-}$). For all investigated systems, a Gaussian broadening with a 75 nm FWHM is applied to $S_0 \rightarrow S_1$ absorption, while a Gaussian broadening with a 125 nm FWHM is applied other absorption processes ($S_1 \rightarrow S_n$, $T_1 \rightarrow T_n$, and $D_0^{+/-} \rightarrow D_n^{+/-}$) due to their broadened nature[6]. Thirdly, we compute the rate of the ISC process from $S_1$ to triplets. Finally, we calculate the spin–orbit coupling (SOC) and the adiabatic energy gap between $T_1$ and $S_0$ to evaluate the lifetime of $T_1$ for these 12 candidates. The radiative decay from $T_1$ is neglected, since no phosphorescent effect has been reported for these materials. Explicit numbers of estimated $\sigma_{em}$ and $\sigma_{abs}^{X_i \rightarrow X_j}$ for 12 selected materials are listed in Table 3, and theoretical rate constants, i.e., the ISC rate from $S_1$ to triplets ($k_{isc}$), the radiative and internal conversion rate ($k_r$ and $k_{ic}$) of $S_1$, are given in Table 4. Good candidates for electrical pumping laser are then screened out as the one that acquires large stimulated cross sections with minimal absorption and annihilation losses, marginal ISC process from $S_1$ to triplets, and a relatively short lifetime $T_1$.

---

**Table 2 π-π interaction energies and dispersion energies of all investigated π-stacking materials[a].**

| Molecule | $E_{tot}$ | $E_{disp}$ | $E_{tot}$ per π-electron | Laser/ASE | Refs. |
|----------|-----------|------------|--------------------------|-----------|-------|
| oMDSB | −12.75 | −18.71 | −0.58 | Yes | 31 |
| CNMODSB | −16.16 | −25.03 | −0.54 | Yes | 54 |
| MSMODSB | −17.37 | −23.10 | −0.58 | Yes | 56 |
| BMSA | −19.90 | −27.57 | −0.66 | Yes | 60 |
| αMODCS | −10.93 | −21.95 | −0.36 | Yes | 23 |
| βMODCS | −21.80 | −33.38 | −0.73 | No | 23 |
| αDBDCS | −22.05 | −30.98 | −0.74 | No | 43 |
| αMODBDCS | −31.81 | −51.55 | −0.94 | No | 23 |
| βMODBDCS | −32.24 | −56.01 | −0.95 | No | 23 |
| βTFDCS | −19.99 | −31.88 | −0.77 | No | 24,29 |
| TPDSB | −18.64 | −26.05 | −0.49 | Yes | 59 |
| TPB | −17.33 | −23.98 | −0.62 | Yes | 65 |

[a]All energetics are shown in the unit of KCal mol$^{-1}$.

---

**Table 3 Values of $S_1$ emission cross section and various absorption cross sections at $\lambda_{em}$ of 12 candidates[a].**

| Molecule | $\sigma_{em}$ | $\sigma_{abs}^{S_0 \rightarrow S_1}$ | $\sigma_{abs}^{S_1 \rightarrow S_n}$ | $\sigma_{abs}^{T_1 \rightarrow T_n}$ | $\sigma_{abs}^{D_0^+ \rightarrow D_n^+}$ | $\sigma_{abs}^{D_0^- \rightarrow D_n^-}$ | $\sigma_{em}^{net,opt}$ | $\sigma_{em}^{net,ele}$ |
|----------|---------------|------|------|------|------|------|------|------|
| oMDSB | 2.67 | 0.26 | 0.10 | 0.25 | 0.48 | 0.49 | 2.31 | 1.09 |
| pMDSB | 3.00 | 0.65 | 0.11 | 0.21 | 0.47 | 0.56 | 2.24 | 1.00 |
| βDCS | 2.31 | 0.22 | 0.06 | 0.54 | 0.42 | 0.86 | 2.03 | 0.21 |
| βDBDCS | 2.76 | 0.31 | 0.10 | 0.33 | 0.28 | 1.39 | 2.35 | 0.35 |
| βPDCS | 1.75 | 0.05 | 0.10 | 0.00 | 0.39 | 0.51 | 1.60 | 0.75 |
| BP3T | 4.49 | 0.04 | 0.00 | 0.18 | 0.76 | 0.30 | 4.45 | 3.21 |
| TPD | 1.22 | 0.20 | 0.05 | 0.14 | 0.62 | 0.61 | 0.97 | NA[b] |
| CzPSBF | 2.20 | 0.25 | 0.00 | 0.06 | 0.36 | 1.00 | 1.95 | 0.53 |
| CzPVSBF | 5.30 | 0.72 | 0.00 | 0.24 | 0.56 | 1.15 | 4.58 | 2.63 |
| TPASBF | 2.90 | 0.30 | 0.05 | 0.12 | 0.70 | 1.23 | 2.55 | 0.50 |
| TSBF | 3.22 | 0.22 | 0.00 | 0.02 | 0.66 | 0.65 | 3.00 | 1.67 |
| BSBCz | 5.34 | 0.25 | 0.00 | 0.19 | 0.61 | 1.24 | 5.09 | 3.05 |

[a]Numbers are shown in the unit of $10^{-16}$ cm$^2$.
[b]The estimated $\sigma_{em}^{net,ele}$ is negative.

**Table 4 Theoretical photophysical properties and experimental threshold power for 12 candidates.**

| Molecule | $k_{isc}$ (s$^{-1}$) | $k_r$ (s$^{-1}$) | $k_{ic}$ (s$^{-1}$) | $\tau_{S_1}$ (s)[a] | $\tau_{T_1}$ (s)[b] | Exp. thresh. (KW cm$^{-2}$) | Refs. |
|---|---|---|---|---|---|---|---|
| oMDSB | $1.5 \times 10^3$ | $5.6 \times 10^8$ | $4.6 \times 10^7$ | $1.7 \times 10^{-9}$ | – | 28 | 31 |
| pMDSB | $9.9 \times 10^2$ | $6.7 \times 10^8$ | $9.6 \times 10^7$ | $1.3 \times 10^{-9}$ | – | 50 | 31 |
| βDCS | $1.2 \times 10^4$ | $4.0 \times 10^8$ | $2.0 \times 10^8$ | $1.7 \times 10^{-9}$ | – | 36 | 62 |
| βDBDCS | $6.6 \times 10^3$ | $3.4 \times 10^8$ | $4.2 \times 10^7$ | $2.7 \times 10^{-9}$ | – | 1.3 | 51 |
| βPDCS | $7.2 \times 10^6$ | $3.3 \times 10^8$ | $4.6 \times 10^7$ | $2.6 \times 10^{-9}$ | – | 23 | 52 |
| BP3T | $2.2 \times 10^6$ | $4.3 \times 10^8$ | $2.3 \times 10^8$ | $1.5 \times 10^{-9}$ | $2.5 \times 10^{-5}$ | 16 | 63 |
| TPD | $1.6 \times 10^6$ | $2.9 \times 10^8$ | $3.7 \times 10^8$ | $1.5 \times 10^{-9}$ | $3.7 \times 10^{-3}$ | 4.2 | 67 |
| CzPSBF | $6.4 \times 10^6$ | $5.1 \times 10^8$ | $6.2 \times 10^8$ | $8.8 \times 10^{-10}$ | $1.2 \times 10^{-4}$ | 5.2 | 66 |
| CzPVSBF | $2.7 \times 10^5$ | $6.3 \times 10^8$ | $3.8 \times 10^8$ | $9.9 \times 10^{-10}$ | $5.1 \times 10^{-5}$ | 0.98 | 7 |
| TPASBF | $5.3 \times 10^5$ | $4.6 \times 10^8$ | $6.1 \times 10^8$ | $9.3 \times 10^{-10}$ | $6.9 \times 10^{-5}$ | 2.4 | 66 |
| TSBF | $2.7 \times 10^5$ | $7.1 \times 10^8$ | $5.8 \times 10^7$ | $1.3 \times 10^{-9}$ | $1.0 \times 10^{-3}$ | 2.2 | 68 |
| BSBCz | $6.6 \times 10^4$ | $6.5 \times 10^8$ | $1.5 \times 10^8$ | $1.3 \times 10^{-9}$ | $1.1 \times 10^{-4}$ | 0.82 | 8 |

[a]$\tau_{S_1}$ is evaluated via $\tau_{S_1} = 1/(k_r + k_{ic} + k_{isc})$.
[b]$\tau_{T_1}$ is evaluated for $T_1/S_0$ SOC > 0.05 cm$^{-1}$ under the assumption that the phosphorescence is negligible. Corresponding SOC and $T_1/S_0$ adiabatic energy gaps are given in Supplementary Table 6.

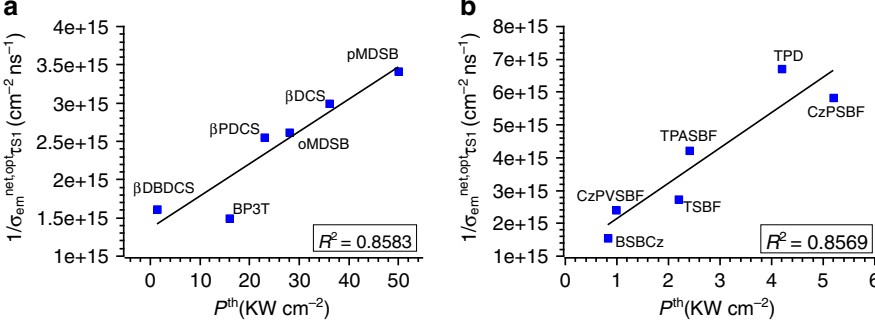

**Fig. 4 Experimental optical pumping threshold power $P^{th}$ with respect to calculated $1/[\sigma_{em}^{net,opt}(\lambda_{em}) \times \tau_{S_1}]$. a** Linear relationship between $P^{th}$ and $1/[\sigma_{em}^{net,opt}(\lambda_{em}) \times \tau_{S_1}]$ for single-crystal materials, and **b** the same linear relationship for thin film materials. The linear fitting coefficient $R^2$ is ~0.85 for both groups, indicating a good linear relationship and thus verifies our computational protocol.

The rate constant calculations are justified by comparing the theoretical predicted PLQY ($\Phi = k_r/[k_r + k_{ic} + k_{isc}]$) with experimental results (Supplementary Table 4), and it can be seen that theoretically predicted values are in good agreement with experiments. To validate the screening algorithm, we first investigate the optical pumping laser performance of these selected systems, where the role played by polaron-induced losses is not significant. For these systems under a pulsed optical pumping source, the triplet–triplet absorption is also negligible since there is no prominent ISC processes from $S_1$ to triplets, which is demonstrated by the fact that $k_{isc}$ is much smaller than $k_r$ and $k_{ic}$ of $S_1$ for all investigated systems as listed in Table 4. We define the net stimulated emission cross section under optical pumping as

$$\sigma_{em}^{net,opt} = \sigma_{em} - \sigma_{abs}^{S_0 \to S_1} - \sigma_{abs}^{S_1 \to S_n}. \quad (4)$$

According to Eq. (4), we compute the value of $\sigma_{em}^{net,opt}$ at the emission wavelength for 12 candidates as listed in Table 3. The computed optical net stimulated emission cross sections match with the magnitude of experimentally determined values, e.g., the experimental stimulated emission cross section for TPASBF is $1.0 \times 10^{-16}$ cm$^{-2}$ according to ref. [34], which rationalizes our computational model for cross sections. For optical pumping lasers, the threshold power $P^{th}$ is inversely proportional to the product of the lifetime of $S_1$ and the effective stimulated emission cross section at the lasing wavelength[35]. Longer $S_1$ lifetime favors occupation number inversion, which is a necessity for conventional lasing mechanism. Practically, we plot the experimental

threshold power with respect to $1/[\sigma_{em}^{net,opt}(\lambda_{em}) \times \tau_{S_1}]$ for both single-crystal materials and thin film materials (Fig. 4), where $\tau_{S_1}$ is evaluated via $\tau_{S_1} = 1/[k_r + k_{ic} + k_{isc}]$ (with explicit numbers listed in Table 4). A linear fitting is performed for both cases, and the linear fitting coefficient $R^2$ is 0.8583 for single-crystal materials and 0.8569 for thin film materials, indicating a convincing linear relationship between experimental $P^{th}$ and theoretically computed $1/[\sigma_{em}^{net,opt}(\lambda_{em}) \times \tau_{S_1}]$. Note that the experimental $P^{th}$ of these selected systems are measured under similar pumping source and pulse width (see references in Table 1 for corresponding experimental conditions); therefore, a good linear relationship can be established, and our screening algorithm is thus verified for optical pumping systems.

The next validating step is to compare the computational triplet–triplet absorption and polaron absorption with available experimental data. Due to the limited experimental data, we first compute the triplet–triplet absorption cross section of a historically prominent system Alq$_3$ and plot it with respect to wavelength in Supplementary Fig. 1a. The absorption peak accurately matches with the experimental spectra, which can be found in ref. [36]. More importantly, we also compute the emission spectra of DCM molecule (which also matches with the experimental data, i.e., 590 nm according to ref. [37]) and it can be seen from Supplementary Fig. 1a that the triplet–triplet absorption spectra of Alq$_3$ significantly overlaps with the emission spectra of DCM, which will not cause severe problem to optically pumped lasing of the host–guest Alq$_3$:DCM system, but will potentially quench the electrically pumped lasing due to

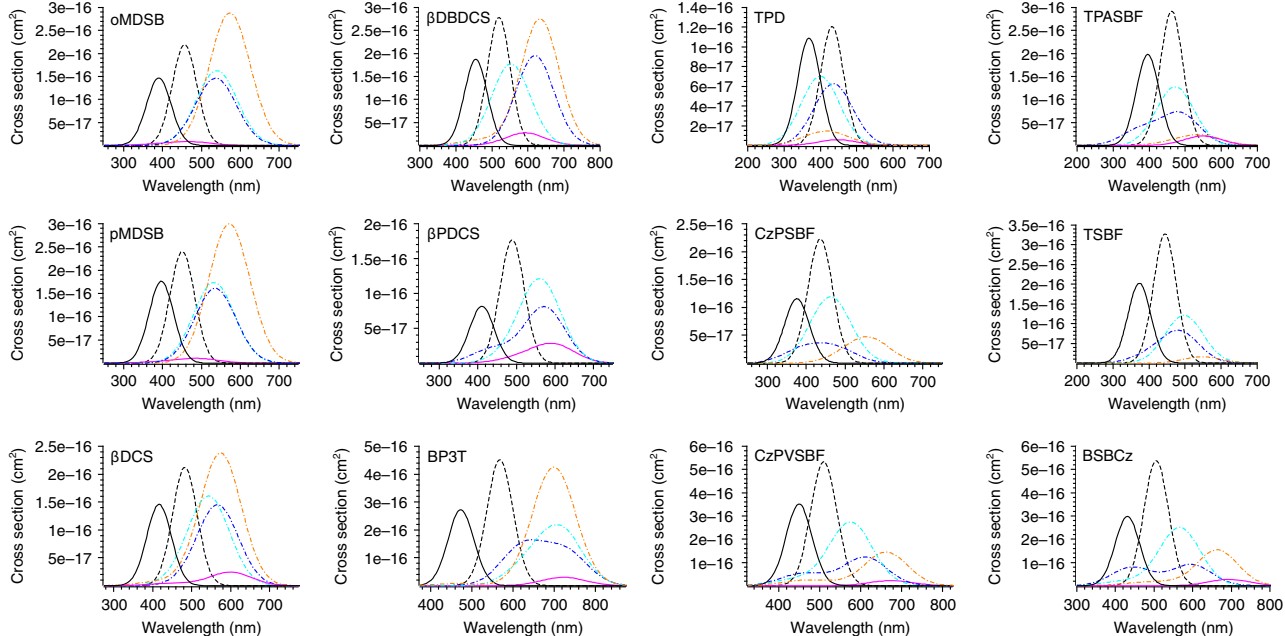

**Fig. 5 Theoretically predicted $S_1$ emission cross sections and various absorption cross sections.** Black solid line and black dashed line denote the self-absorption cross section and stimulated emission cross section of $S_1$, respectively. Various absorption cross sections (if any) introduced by $S_1 \rightarrow S_n$ (magenta solid), $T_1 \rightarrow T_n$ (orange dot-dashed), $D_0^+ \rightarrow D_n^+$ (blue dot-dashed), and $D_0^- \rightarrow D_n^-$ (cyan dot-dashed) are shown in the vicinity of the emission wavelength (±125 nm). Losses under optical pumping are indicated by solid lines, which are not significant for all candidates, while losses under electrical pumping are indicated by both solid and dot-dashed lines. Most candidates suffer from losses introduced by triplet excitons and polarons as indicated by the value of $\sigma_{abs}^{T_1 \rightarrow T_n}$, $\sigma_{abs}^{D_0^+ \rightarrow D_n^+}$, and $\sigma_{abs}^{D_0^- \rightarrow D_n^-}$ at the emission wavelength, while such losses are moderate compared to the large stimulated emission cross section of $S_1$ for BP3T, CzPVSBF, and BSBCz.

the large amount of Alq$_3$ triplet excitons under current injection. Therefore, the electrical pumping laser is unlikely to be realized on Alq$_3$:DCM system. The computational polaron absorption is examined for TPD cation in Supplementary Fig. 1b. The overall line shape as well as the two main peaks are in good agreement with experimental results, i.e., 484 nm and 1400 nm, respectively according to ref. [38]. Altogether, the rationality of our computational method and screening protocol are primitively justified.

To screen out good candidates for electrically pumped lasing, in Fig. 5, we plot the stimulated emission cross section of $S_1$ and various abovementioned absorption cross sections (if any) in the vicinity of the emission wavelength (±125 nm, with the corresponding energy range ±0.5–1 eV). Explicit numbers for emission and absorption cross section at the emission wavelength are given in Table 3, while explicit transition energies and corresponding oscillator strengths can be found in Supplementary Table 5. According to Fig. 5 and the investigation for optical pumping case, these 12 materials do not significantly suffer from $S_0 \rightarrow S_1$ self-absorption nor singlet exciton-induced losses. These findings are consistent with the fact that the chosen materials are experimentally excellent laser gain media with low laser/ASE thresholds under optical pumping. Nevertheless, most of the selected materials suffer from polaron absorptions (blue and cyan dot-dashed lines in Fig. 5), and triplet–triplet absorption (orange dot-dashed line in Fig. 5) cannot be ignored for electrical pumping because of the large population for triplet exciton formed under current injection. Among these materials, BP3T, CzPVSBF, and BSBCz have larger stimulated emission cross sections than other molecules, and various absorption cross sections at the emission wavelength are small compared to the value of $\sigma_{em}$, indicating a potentially large net emission cross section for electrical pumping, which is intuitively estimated in

Table 3 as

$$\sigma_{em}^{net,ele} = \sigma_{em} - \sigma_{abs}^{S_0 \rightarrow S_1} - \sigma_{abs}^{S_1 \rightarrow S_n} - \sigma_{abs}^{T_1 \rightarrow T_n} - \sigma_{abs}^{D_0^+ \rightarrow D_n^+} - \sigma_{abs}^{D_0^- \rightarrow D_n^-}.$$

(5)

Furthermore, these three materials have short $T_1$ lifetimes (Table 4), which will hinder the pileup of triplet excitons under current injection. Note that selected single-crystal materials other than BP3T have almost zero SOC between $T_1$ and $S_0$ from Supplementary Table 6, which will lengthen the lifetime of $T_1$ and thus aggravate the triplet-induced losses and may lead to the thermal degradation of the material. For thin film materials, TPD, TPASBF, and CzPSBF have significantly larger polaron absorption at the emission wavelength, which excludes them from good candidates for electrical pumping laser. TSBF has relatively milder polaron-induced losses, but its $\sigma_{em}^{net,ele}$ is still significantly smaller compared to CzPVSBF and BSBCz (Table 3), with a relatively long triplet lifetime (Table 4). Therefore, based on the above screening process, we conclude that BP3T, CzPVSBF, and BSBCz have the greatest potential in realizing electrically pumped laser among these selected systems.

Finally, a good electrical pumping material should have a relatively high mobility for both electron and hole. Theoretically predicted hole and electron mobilities for BP3T is 0.74 cm$^2$ V$^{-1}$ S$^{-1}$ and 0.59 cm$^2$ V$^{-1}$ S$^{-1}$, respectively, which are in good agreement with experimental values, i.e., 1.64 cm$^2$ V$^{-1}$ S$^{-1}$ and 0.17 cm$^2$ V$^{-1}$ S$^{-1}$ (ref. [39]). Due to the fact that electrons are more easily trapped in practice[40], the predicted electron mobility is larger than the experimental values. In spite of this, the computational results reveal a good capability for both electron and hole transporting in BP3T. Since the mobility calculation is nontrivial for discorded systems, we implicitly estimate the mobility for CzPVSBF and BSBCz by calculating the anion and cation reorganization energies (Table 5).

**Table 5 Hole/electron mobility and cation/anion reorganization energy for BP3T, CzPVSBF, and BSBCz.**

| Molecule | Hole mobility | Electron mobility | Cation reorganization energy | Anion reorganization energy |
|---|---|---|---|---|
| BP3T | 0.74 $cm^2\,Vs^{-1}$ | 0.59 $cm^2\,Vs^{-1}$ | 303 meV | 424 meV |
| CzPVSBF | NA | NA | 229 meV | 281 meV |
| BSBCz | NA | NA | 149 meV | 337 meV |

In principle, smaller reorganization energies for both anion and cation are preferred during the charge transport process. It can be seen from Table 5 that the anion and cation reorganization energies of CzPVSBF are very close, and the values are even smaller than BP3T, suggesting a presumably balanced mobility of electrons and holes. For BSBCz, the anion reorganization is twice as large as the cation reorganization energy, which is only because the cation reorganization for BSBCz is extremely small, indicating a potentially superior hole mobility and a relatively good electron mobility. In fact, light narrowing has been observed under electrical pumping from BSBCz with a sophisticatedly designed distributed feedback structure[10], which verifies the theoretical prediction we present here.

## Discussion

In conclusion, employing theoretical approaches, including DFT/ TDDFT for electronic structure and MOMAP for the molecular photophysical parameters, we have investigated the influence of the emission oscillator strength and the intermolecular π–π interaction to the general lasing behavior over a wide range of organic fluorescent materials, followed by a systematic screening of electrically pumped lasing candidates over 12 molecules with low optical pumping laser/ASE threshold based on the criteria that good candidate should have a large stimulated emission cross section and simultaneously avoid the existence of any strong absorption/annihilation processes induced by excitons and polarons near the $S_1$ emission wavelength, which is checked via the absorption cross sections of $S_0 \rightarrow S_1$, $S_1 \rightarrow S_n$, $T_1 \rightarrow T_n$, and $D_0^{+/-} \rightarrow D_n^{+/-}$. The reliability of the theory is embodied not only by the calculated excited state energy levels in good agreement with the measurements, but also by the strong correlation between the experimental optical pumping threshold power with the calculated inverse product of net emission cross section with $S_1$ lifetime, as it should be for optically pumped lasing action. The ISC loss from $S_1$ to triplets, as well as the accumulation of $T_1$ under electrical pumping should also be avoided, which are checked by comparing the ISC rate and the radiation rate of $S_1$ and evaluating the lifetime of $T_1$ based on $T_1/S_0$ SOC. Moreover, relatively high mobilities for both electrons and holes are also preferred for good electrical pumping candidates. With these criteria and the photophysical parameters obtained from MOMAP[11–13], three promising candidates (i.e., BP3T, BSBCz, and CzPVSBF) have been predicted with great potential in realizing electrically pumped lasing. All of these three materials have a short lifetime of $T_1$, and a relatively slow ISC rate compared to the radiation rate of $S_1$. Compared to the large stimulated emission cross section of these three molecules, no significant absorption cross section of any type has been predicted. Furthermore, computational results, i.e., mobilities of electron and holes for BP3T, and reorganization energy of polarons for CzPVSBF and BSBCz, demonstrate a potentially good capability of electron and hole transporting of these three materials.

While this manuscript is being prepared, new organic gain materials with low laser/ASE thresholds continuously spring up. We believe the computational strategy we have presented in this paper can serve as a general protocol for molecular design of lasing materials in the future, especially for the machine-learning-based research that would require a number of molecular descriptors as presented in this work.

## Methods

**Electronic structure calculation**. Unless otherwise specified, B3LYP functional and 6-31G(d) basis set are applied to all calculations throughout this work. The singlet ground states are optimized via a normal restricted DFT calculation, while the geometry and the excitation energy of singlet excited states are obtained from restricted TDDFT. The geometries of the $T_1$ states are optimized using an unrestricted version DFT method, and triplet excitation energies are then calculated via TDDFT. Polarons with either positive or negative charge are treated as cations or anions, respectively, via unrestricted DFT/TDDFT. B3LYP functional is often considered as a reliable descriptor in predicting the excitation energy for solid-state materials because it enjoys an error cancelation between the underestimated fundamental gap and the absence of renormalization[41]. While B3LYP functional gives reasonably accurate results for excited states with local excitation character, it always fails to provide a reliable prediction to CT excited states[41,42]. Therefore, an optimal-tuning version[41] of LRC-ωPBE functional is applied to validate the CT excited states properties predicted by B3LYP. All electronic structure calculations are performed on single-molecule (gas phase) models, with an overall MAE < 0.1 eV for excited state energies of all investigated materials. In fact, the single-molecule model has been widely employed to analyze both the molecular orbitals and the excitation energies of solid-state organic PL systems[43–45]. We also take one organic single-crystal material (βDBDCS) as an example and show in Supplementary Fig. 2, and Supplementary Tables 7 and 8 that the optimized structures and excitation energetics obtained from quantum mechanical and molecular mechanical calculation do not significantly differ from those predicted by single-molecule calculations. The intermolecular π–π interaction in crystals is addressed between the nearest dimers via the XSAPT + MBD method proposed by Herbert et al.[25]. A monomer with its nearest neighbor are directly taken from the crystal cif file and form a dimer for a given material. Since the dispersion energy evaluated by MBD is found to be sensitive to the choice of basis set[25], the def2-tzvpp basis set is applied for this part to get a more accurate description. All quantum chemistry calculations are performed in quantum chemistry package Q-Chem 5.2 (ref. [46]) except for the frequency analysis, which are performed in Gaussian 16 package[47].

**Photophysical properties calculation**. All rate constant calculations and mobility calculations are performed via thermal vibration correlation function (TVCF) method in MOMAP 2019B[11–13], which has been successfully applied in a wide range to predict various optoelectrical properties of organic molecules[48,49]. During the evaluation of the internal conversion rate constant $k_{ic}$, we have applied a Lorentzian-type broadening (FWHM = 200 $cm^{-1}$) in time domain when TVCF fails to converge. For mobility calculation, we have applied B3LYP functional with 6-31G(d) basis set for frequency analysis and PW91PW91 functional with 6-31G(d) basis set for the computation of transfer integral according to ref. [50].

## Data availability

The authors declare that the necessary data supporting the findings of this study are available within the paper (and its Supplementary Information files). All data are available from the corresponding author upon reasonable request.

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

## Acknowledgements

This work was supported by the National Natural Science Foundation of China through the project "Science Center for Luminescence from Molecular Aggregates (SCELMA)," Grant No. 21788102, as well as by the Ministry of Science and Technology of China through the National Key R&D Plan, Grant No. 2017YFA0204501. Q.O. thank Dr. Yongsheng Zhao, Dr. Yongli Yan, and Dr. Shan Mei for inspiring discussions. Q.O. is also supported by the Shuimu Tsinghua Scholar Program.

## Author contributions

Z.S. conceived the project. Q.O. carried out the calculations. Q.P. helped with the calculation process and provided inspiring suggestions for improvement. Q.O., Q.P., and Z.S. all contributed to data analysis and writing the paper.

## Competing interests

The authors declare no competing interests.
