## [Peer Review File · Nature Communications]

Reviewers' Comments:

Reviewer #1:

Remarks to the Author:

In this manuscript, Shuai et al. reported a systematically theoretical study to screen out electrical pumping lasing molecules over a wide range of organic fluorescent 14 molecules. The emission energies, emission oscillator strength, n-n interaction energies, the ISC rate, the SOC, and even the lifetime of T1 were computed by performing the tested DFT/TDDFT calculations and then used as the screening toolbox. Several valuable insights were concluded for the rational design of electrical pumping lasing molecules, and three compounds including BP3T, BSBCz, and CzPVSBF were identified as the promising candidates for electrically pumped lasing. In my opinion, the reported computational strategy would be important and should attract general interest in the field of molecular design of organic lasing materials. Therefore, I recommend the publication of the manuscript in Nature Communications after minor revision.

1. In Scheme 1, the font sizes seem to be very different, and the group names are too small to see clearly. I suggest that the authors improved the quality of Schemes and Figures significantly for the publication.

2. Is it necessary to retain three significant digits after the decimal point for the energy unit Kcal/mol? I think two digits would be enough accurate for drawing the correct conclusion. In addition, the three digits would be enough accurate for eV.

3. In the Line 122, "(A detailed energy decomposition is listed in Table S5.)" is very strange to be written in the main text, is it a correct sentence? Are the brackets necessary?

4. Some kinds of the grammar mistakes exist in the manuscript. For example, the "we also calculate ..." in Line 123 should be replaced by "we also calculated ...". I suggest the authors carefully revised their manuscript before publication.

5. Actually, I have not seen the description of the "n-n overlap" before, is it n-n stacking? The overlap of the orbitals generally presents the formation of a chemical bond, if it is so or has other meaning, the authors should give the detailed explanation.

6. The format of the unit should be seriously checked and corrected.

Reviewer #2:

Remarks to the Author:

This manuscript reports a computational exploration of molecules for organic laser diodes that satisfy the criteria of having a large stimulated emission cross-section, a short triplet lifetime, and negligible overlap with polaron/excited state absorption bands. The authors explore a large number of molecules and identify three particularly promising candidates, one of which (BSBCz) is the subject of a current experimental laser diode claim. I think the overall aim of the manuscript is fantastic and sorely needed in a field largely dominated by experimental trial and error.

My main concern with this work is how good a predictive tool it really is as far as lasing goes. Certainly the authors have tried to make the connection to experiment in showing good agreement between the computed and measured emission energies. But the whole point of this manuscript is that many other factors go into establishing the quality of a lasing material and so what would have been really nice to see is a direct correlation of the quantities calculated in the manuscript with some experimental measure of lasing. For example some sort of scatter plot between the measured ASE thresholds and some collection of calculated parameters or figure of merit (e.g. stimulated emission cross-section*Stokes shift/singlet excited state absorption or something like that) that demonstrated a correlation. I don't know if the authors can do this, but I think it would strengthen the manuscript considerably.

A few other comments that would be helpful to address:

1. The Stokes shift is a relevant parameter that should be calculated since these molecules are all

4-level laser systems. Having a minimal overlap with the ground state absorption is thus a key factor in minimizing the threshold, as reflected by the common strategy of doping emitters into a wide gap host to reduce the transparency threshold.

2. Another area where it would be really valuable to see a connection with experiment is in the predicted energies & oscillator strengths of the polaron/excited state absorption bands. This information is admittedly harder to come by in the experimental literature, but if the authors could find it for at least a couple of the molecules to show agreement, I think it would be valuable. For example, the TPD cation absorption is given in *J. Mater. Chem.*, 2005, 15, 2304–2315.

3. Giving some sense of the breadth of the absorption bands associated with the transitions shown in Fig. 2 would be valuable. Many polaron and triplet absorption bands are hundreds of nm wide, which makes the +/-50 nm criterion for inclusion somewhat questionable.

4. Given the large amount of work on and historical prominence of the host-guest system Alq3:DCM, it would have been nice to include this as a reference point - if only to show why this system has not worked for electrically pumped lasing due, e.g. to some overlapping excited state/polaron absorption band.

5. The authors do a good job considering the factors that impact the threshold density of an electrically-pumped laser, but there is nothing about the electrical considerations of the device - in particular the desire to have balanced e⁻ and h⁺ mobilities that are, ideally high to minimize the drive voltage of the device. Predicting mobilities is notoriously difficult from computation, so it may not be possible to be quantitative here but there should at least be some discussion of the electrical properties that factor in to a promising candidate material.

Reviewer #3:

None

Response to referees:

Reviewer #1 (Remarks to the Author):

In this manuscript, Shuai et al. reported a systematically theoretical study to screen out electrical pumping lasing molecules over a wide range of organic fluorescent 14 molecules. The emission energies, emission oscillator strength, π - π interaction energies, the ISC rate, the SOC, and even the lifetime of T1 were computed by performing the tested DFT/TDDFT calculations and then used as the screening toolbox. Several valuable insights were concluded for the rational design of electrical pumping lasing molecules, and three compounds including BP3T, BSBCz, and CzPVSBF were identified as the promising candidates for electrically pumped lasing. In my opinion, the reported computational strategy would be important and should attract general interest in the field of molecular design of organic lasing materials. Therefore, I recommend the publication of the manuscript in Nature Communications after minor revision.

1. In Scheme 1, the font sizes seem to be very different, and the group names are too small to see clearly. I suggest that the authors improved the quality of Schemes and Figures significantly for the publication.

Author's response: We thank the reviewer for the suggestion. The molecular structures in Scheme 1 and Scheme 2 have been remade with higher resolution and the fonts have been adjusted. Other figures have also been remade with improved quality.

2. Is it necessary to retain three significant digits after the decimal point for the energy unit Kcal/mol? I think two digits would be enough accurate for drawing the correct conclusion. In addition, the three digits would be enough accurate for eV.

Author's response: We thank the reviewer for the suggestion. We have now kept two digits for the energy unit KCal/mol and three digits for eV.

3. In the Line 122, "(A detailed energy decomposition is listed in Table S5.)" is very strange to be written in the main text, is it a correct sentence? Are the brackets necessary?

Author's response: We thank the reviewer and the parentheses have been removed. This sentence is grammatically correct though.

4. Some kinds of the grammar mistakes exist in the manuscript. For example, the "we also calculate ..." in Line 123 should be replaced by "we also calculated ...". I suggest the authors carefully revised their manuscript before publication.

Author's response: We thank the reviewer's suggestions and we have changed "we also calculate" in Line 123 to "we have also calculated". In addition, the language of the whole manuscript has been carefully checked.

5. Actually, I have not seen the description of the " π - π overlap" before, is it π - π stacking? The overlap of the orbitals generally presents the formation of a chemical bond, if it is so or has other meaning, the authors should give the detailed explanation.

Author's response: We thank the reviewer's suggestion. Actually π - π overlap in this manuscript shares the same definition as π - π interaction. For rigorousness, we have changed π - π overlap to π - π interaction throughout the manuscript.

6. The format of the unit should be seriously checked and corrected.

Author's response: We thank the reviewer's suggestion. The format of the unit throughout the manuscript has been carefully checked.

Reviewer #2 (Remarks to the Author):

This manuscript reports a computational exploration of molecules for organic laser diodes that satisfy the criteria of having a large stimulated emission cross-section, a short triplet lifetime, and negligible overlap with polaron/excited state absorption bands. The authors explore a large number of molecules and identify three particularly promising candidates, one of which (BSBCz) is the subject of a current experimental laser diode claim. I think the overall aim of the manuscript is fantastic and sorely needed in a field largely dominated by experimental trial and error.

My main concern with this work is how good a predictive tool it really is as far as lasing goes.

Certainly the authors have tried to make the connection to experiment in showing good agreement between the computed and measured emission energies. But the whole point of this manuscript is that many other factors go into establishing the quality of a lasing material and so what would have been really nice to see is a direct correlation of the quantities calculated in the manuscript with some experimental measure of lasing. For example some sort of scatter plot between the measured ASE thresholds and some collection of calculated parameters or figure of merit (e.g. stimulated emission cross-section*Stokes shift/singlet excited state absorption or something like that) that demonstrated a correlation. I don't know if the authors can do this, but I think it would strengthen the manuscript considerably.

Author's response: We deeply appreciate the reviewer's important suggestion. Now we have changed the screening parameter from absorption/emission oscillator strength to absorption/emission cross section. In addition, we have made two scatter plots between the optical pumping threshold power (P^{th}) and the reciprocal of the product of theoretical net stimulated emission cross section under optical pumping and theoretical lifetime of S_1 ($1/[\sigma_{\text{emi}}^{\text{net,opt}}(\lambda_{\text{emi}}) \cdot \tau_{S_1}]$) for selected single crystal and thin film materials, respectively (Figure 2). In principle, P^{th} is inversely proportional to $\sigma_{\text{emi}}^{\text{net,opt}}(\lambda_{\text{emi}}) \cdot \tau_{S_1}$ (Laporta and Brussard, 1991), which has been successfully reproduced by our scatter plot. The linear fitting coefficients between experimental P^{th} and theoretical $1/[\sigma_{\text{emi}}^{\text{net,opt}}(\lambda_{\text{emi}}) \cdot \tau_{S_1}]$ are around 0.85 for both single crystal and thin film materials respectively based on our calculation, which further rationalizes our computational protocol.

A few other comments that would be helpful to address:

1. The Stokes shift is a relevant parameter that should be calculated since these molecules are all 4-level laser systems. Having a minimal overlap with the ground state absorption is thus a key factor in minimizing the threshold, as reflected by the common strategy of doping emitters into a wide gap host to reduce the transparency threshold.

Author's response: We thank the reviewer's suggestion and we have reported the theoretical reorganization energy of all investigated materials in Table 2, which is approximately the Stokes shift of each material. In addition, we have addressed the influence of self-absorption by evaluating the value of S_1 absorption cross section at its emission wavelength.

2. Another area where it would be really valuable to see a connection with experiment is in the predicted energies & oscillator strengths of the polaron/excited state absorption bands. This information is admittedly harder to come by in the experimental literature, but if the authors could find it for at least a couple of the molecules to show agreement, I think it would be valuable. For example, the TPD cation absorption is given in J. Mater. Chem., 2005, 15, 2304–2315.

Author's response: We thank the reviewer's suggestion. We have plotted the triplet-triplet absorption cross section for Alq₃ and the cation absorption cross section for TPD, and the theoretical results (Figure 3(a) and Figure 3(b)) have been compared with existing experimental spectra. The overall lineshape and the position of the main peaks are in good agreement with experimental results.

3. Giving some sense of the breadth of the absorption bands associated with the transitions shown in Fig. 2 would be valuable. Many polaron and triplet absorption bands are hundreds of nm wide, which makes the +/-50 nm criterion for inclusion somewhat questionable.

Author's response: We thank the reviewer's suggestion. We have now expanded the region to ± 125 nm.

4. Given the large amount of work on and historical prominence of the host-guest system Alq₃:DCM, it would have been nice to include this as a reference point - if only to show why this system has not worked for electrically pumped lasing due, e.g. to some overlapping excited state/polaron absorption band.

Author's response: We thank the reviewer's suggestion. We have analyzed this system in Figure 3(a), and our prediction is that the triplet-triplet absorption of Alq₃ significantly overlaps with the emission of DCM molecule, which will not cause severe problem under optical pumping where the population of Alq₃ triplet exciton is limited, but will presumably quench the lasing action of DCM under electrical pumping where the population of Alq₃ triplet exciton is much larger, resulting an aggravated triplet-induced loss.

5. The authors do a good job considering the factors that impact the threshold density of an electrically-pumped laser, but there is nothing about the electrical considerations of the device - in particular the desire to have balanced e- and h+ mobilities that are, ideally high to minimize the drive voltage of the device. Predicting mobilities is notoriously difficult from computation, so it may not be possible to be quantitative here but there should at least be some discussion of the electrical properties that factor in to a promising candidate material.

Author's response: We thank the reviewer's suggestion. We have computed the hole and electron mobilities for BP3T, which are $0.74 \text{ cm}^2\text{V}^{-1}\text{S}^{-1}$ and $0.59 \text{ cm}^2\text{V}^{-1}\text{S}^{-1}$, respectively. These numbers match with the magnitude of the experimental values, revealing a good capability for both electron and hole transporting in BP3T. While it is nontrivial to compute the mobility for disordered materials, we have computed the anion and cation reorganization energies for CzPVSBF and BSBCz (Table 6). In principle, a small reorganization energy is preferred for materials with high mobilities. Both the anion and cation reorganization energies for CzPVSBF and BSBCz are relatively small (equal or less than 300meV), indicating a potentially good capability for electron and hole transporting of these two materials.

Reviewers' Comments:

Reviewer #1:

Remarks to the Author:

I have carefully checked the point-by-point responses to the reviewer's questions and suggestions, and I am satisfied for the revision, so I recommend the publication of the revised manuscript now.

Reviewer #2:

Remarks to the Author:

The authors have addressed all my comments. I recommend this manuscript be published in its current form.

Response to referees:

Reviewer #1 (Remarks to the Author):

I have carefully checked the point-by-point responses to the reviewer's questions and suggestions, and I am satisfied for the revision, so I recommend the publication of the revised manuscript now.

Author's response: we sincerely appreciate the careful reading and all suggestions provided by the reviewer.

Reviewer #2 (Remarks to the Author):

The authors have addressed all my comments. I recommend this manuscript be published in its current form.

Author's response: we sincerely appreciate the careful reading and all suggestions provided by the reviewer.